# Corrosion Behavior of CrFeCoNiV_x_ (x = 0.5 and 1) High-Entropy Alloys in 1M Sulfuric Acid and 1M Hydrochloric Acid Solutions

**DOI:** 10.3390/ma15103639

**Published:** 2022-05-19

**Authors:** Chun-Huei Tsau, Jo-Yi Chen, Tien-Yu Chien

**Affiliations:** Institute of Nanomaterials, Chinese Culture University, Taipei 111, Taiwan; zoe85122@yahoo.com.tw (J.-Y.C.); school0952067023@gmail.com (T.-Y.C.)

**Keywords:** CrFeCoNiV_x_, microstructure, corrosion, potentiodynamic polarization test, electrochemical impedance spectroscopy

## Abstract

CrFeCoNiV_x_ high-entropy alloys were prepared by arc-melting, and the microstructures and corrosion properties of these alloys were studied. The CrFeCoNiV_0.5_ alloy had a granular structure; the matrix was a face-centered cubic (FCC) structure, and the second phase was a σ phase with a tetragonal structure. The CrFeCoNiV alloy had a dendritic structure; the dendrites in this alloy showed an FCC phase, and the interdendrities had a eutectic structure of FCC and σ phases. Therefore, CrFeCoNiV was much harder than the CrFeCoNiV_0.5_ alloy due to the dendritic structures. The potentiodynamic polarization test and electrochemical impedance spectroscopy were used to evaluate the corrosion behavior of the CrFeCoNiV_x_ high-entropy alloys in deaerated 1M sulfuric acid and 1M hydrochloric acid solutions. The results indicated that the CrFeCoNiV_0.5_ alloy had a better corrosion resistance because of the granular structure.

## 1. Introduction

The high-entropy alloy (HEA) concept has been studied for many years. Prof. Yeh pointed out that the four major effects of this high-entropy alloy concept were (1) high entropy, (2) sluggish diffusion, (3) severe lattice distortion and (4) cocktail effects [1,2,3]. This high-entropy concept also provides a new way to select suitable elements and develop new alloys for application. Therefore, many new alloy systems have been investigated to improve the mechanical, physical and chemical properties according to the high-entropy alloy concept. To develop refractory alloys, high melting-point elements have been selected to prepare NbMoTaW-, HfTaTiNbZr- and VNbMoTaW-based alloys [4,5]. To improve the mechanical properties, CoCrFeNiTiAl_x_ high-entropy alloys have been studied, and the CoCrFeNiTiAl alloy was found to possess good compressive strength and elastic modulus [6]. The metallurgy, heat treatment and working processes of the CoCrFeMnNi high-entropy alloy have been investigated well to understand the potential of this alloy [7,8]. Additionally, the mechanical alloy process has been used to prepare HEAs with nanocrystalline structures, thus improving the mechanical properties of the alloys [9,10]. Many high-entropy alloys based on the CrFeCoNi-system have been designed for improving the corrosion resistance. Increasing the Al and Cr contents could increase the corrosion resistance of Al_x_CrFeCoNi high-entropy alloys in 3.5% NaCl solution [11]. Adding Mo could improve corrosion resistance of (CoCrFeNi)_100−x_Mo_x_ alloys in 3.5% NaCl solution [12]. CrFeCoNi(Nb,Mo) alloys and CrFeCoNiNb_x_ alloys also showed good corrosion resistance when compared with commercial 304 stainless steel [13,14]. AlCr*_x_*NiCu_0.5_Mo alloys coated on the surface of Q235 alloy steel could significantly improve the corrosion resistance in 3.5% NaCl solution [15]. The NbTiAlSiZrN_x_ alloy coated on the surface of 304 stainless steel could enhance the corrosion resistance in 0.5 M H_2_SO_4_ solution [16].

The pure element of vanadium has good corrosion resistance in phosphoric, sulfuric and hydrochloric acid solutions but poor in nitric acid solution [17]. Adding V into Ti6AlxV alloys showed good corrosion resistance in 3.5 wt.% NaCl solution because vanadium could decrease the corrosion rate by forming oxide films on the surfaces of Ti6AlxV alloys [18]. Therefore, the present work studied the effect of vanadium content on the microstructures of CrFeCoNiV_x_ high-entropy alloys and the corrosion properties of these alloys in deaerated 1M sulfuric acid and hydrochloric acid solutions.

## 2. Materials and Methods

The experimental alloys, CrFeCoNiV_0.5_ and CrFeCoNiV, were prepared by arc-melting in argon atmosphere with a pressure of about 50 kPa (0.5 atm). Table 1 lists the nominal compositions of these alloys. The purity of the five elements, Cr, Fe, Co, Ni and V, were all higher than 99.9%. Each melt had a total weight of about 120 g. The crystal structures of the alloys were examined using an X-ray diffractometer (XRD; Rigagu ME510-FM2), which was operated at 30 kV. A standard metallurgy process was used to prepare the specimens for observing the microstructures of the alloys. The etching solution was aqua regia (three parts of HCl and one part of HNO_3_). The microstructures of the specimens were observed using a scanning electron microscope (SEM; JEOL JSM-6335F), which was operated at 10 kV. The chemical compositions of the alloys and the phases in these alloys were examined using an energy-dispersive spectrometer (SEM/EDS). The hardness of the alloys was measured using a Vickers hardness tester (Matsuzawa MMT-X3B) at a loading force of 19.6 N (2 kgs).

The potentiodynamic polarization curves, electrochemical impedance spectroscopy (EIS) and simulation software of NOVA 2.1.4 of the CoCrFeNiV_x_ alloys were measured using an electrochemical instrument (potentiostat/galvanostat; Autolab PGSTAT302N). This device had three electrodes, which were working specimen, reference and counter electrode. All of the working specimens were mounted in epoxy resin with an exposed area of 0.1964 cm^2^ (0.5 cm in diameter). The surfaces of the specimens for the polarization test were mechanically polished using 1200 SiC grit paper. The saturated silver chloride electrode (Ag/AgCl, V_SSE_) was used as a reference; the reduction potential of a saturated silver chloride electrode is 0.197 V higher than that of the standard hydrogen electrode (V_SHE_) at 25 °C [19]. A smooth Pt sheet was used as the counter electrode. The solutions were deaerated by bubbling N_2_ gas during the polarization test. The scanning rate of the potentiodynamic polarization test was 1MV/s. The solutions were prepared with reagent-grade acids and deionized water.

## 3. Results and Discussion

Figure 1a,b show the SEM microstructures of as-cast CrFeCoNiV_0.5_ and CrFeCoNiV alloys. The CrFeCoNiV_0.5_ alloy showed a granular structure, and some particles (σ phase) were observed in the matrix. On the other hand, the CrFeCoNiV alloy showed a dendritic structures, with the darker areas being the FCC phase and the brighter areas being the σ phase. The dendrites of the CrFeCoNiV alloy showed an FCC phase; the interdendrities of the CrFeCoNiV alloy had a eutectic structure, and the FCC phase was distributed in the interdendrities in a slender shape. Table 2 lists the overall chemical compositions of the alloys and the compositions of the phases in the alloys. The Cr and V contents in the σ phase were higher than those in the FCC phase; the FCC phase had higher Fe, Co and Ni contents. Figure 2 shows the XRD patterns of as-cast CrFeCoNiV_x_ alloys. According to the works [20,21], the two phases in the CrFeCoNiV_x_ alloys were FCC and σ phases, and the σ phase had a tetragonal structure referring to JCPDS 07-0383. The average hardness values of the CrFeCoNiV_0.5_ and CrFeCoNiV alloys were 146 HV and 471 HV, respectively. The large hardness difference between these alloys was also caused by the different structures. The hardness of the tetragonal structure was greater than that of the FCC phase, as is well known, because the slip systems of the FCC phase are more numerous than those of the tetragonal phase.

Figure 3a,b shows the potentiodynamic polarization curves of the CrFeCoNiV_0.5_ and CrFeCoNiV alloys tested in deaerated 1M H_2_SO_4_ solution at 30 and 60 °C, respectively. The important potentiodynamic polarization data of the curves are listed in Table 3. The cathodic polarization curves of the alloys means the potential of the curves negative than corrosion potential (*E*_corr_). On the contrary, the anodic polarization curves of the alloys means the potential of the curves positive than corrosion potential (*E*_corr_). The corrosion potential of the CrFeCoNiV_0.5_ alloy was more positive than that of the CrFeCoNiV alloy; both the corrosion current densities (*i*_corr_) of these two alloys were close at 30 °C. The CrFeCoNiV_0.5_ alloy almost had no passivation potential, but the CrFeCoNiV alloy had a small anodic peak. The values of the passivation potential (*E*_pp_) and critical current density (*i*_crit_) of the anodic peak of the CrFeCoNiV_x_ alloy are listed in Table 3. The passivation current density of the CrFeCoNiV_0.5_ alloy almost kept the same value when the applied potential was less than 0.7 V_SHE_. However, the passivation current density of the CrFeCoNiV alloy changed with the applied potential. The secondary anodic peaks of the CrFeCoNiV_x_ alloy were observed at about 1 V_SHE_. The values of the passivation potential (*E*_pp2_) and critical current density (*i*_crit2_) of the secondary anodic peak of the CrFeCoNiV_x_ alloy are also listed in Table 3. The passivation regions of the CrFeCoNiV_x_ alloys were broken down at a potential (*E*_b_) of about 1.2 V_SHE_ because of the oxygen evolution reaction [22]. The potentiodynamic polarization curves of the CrFeCoNiV_x_ alloys tested in deaerated 1M H_2_SO_4_ solution at 60 °C are shown in Figure 3b. The major differences in the CrFeCoNiV_x_ alloys tested at 30 and 60 °C were the passivation regions of the CrFeCoNiV_0.5_ alloy. At the testing temperature of 60 °C, the passivation current density of the CrFeCoNiV_0.5_ alloy increased with the increase in the applied potential. The passivation region of the CrFeCoNiV_x_ alloys were also broken down at a potential (*E*_b_) of about 1.2 V_SHE_. Moreover, Moreover, the corrosion current densities, critical current densities for the anodic peaks of these alloys increased by comparing with the values tested under 30 °C, because the electrochemical reaction increased with the increase in the testing temperature. At these testing temperatures, the corrosion potentials of the CrFeCoNiV alloy were more negative than those of the CrFeCoNiV_0.5_ alloy; the corrosion current densities and passivation current densities of the CrFeCoNiV alloy were larger than those of the CrFeCoNiV_0.5_ alloy. Therefore, the corrosion resistance of the CrFeCoNiV_0.5_ alloy was better than that of the CrFeCoNiV alloy in deaerated 1M H_2_SO_4_ solution.

The Nyquist plots, Bode plots and equivalent circuit diagrams of the CoCrFeNiV_x_ alloys tested by electrochemical impedance spectroscopy (EIS) in deaerated 1M H_2_SO_4_ solution are shown in Figure 4. The start points of the CoCrFeNiV_x_ alloys in the Nyquist plot were very close, as shown in Figure 4a. The Nyquist curve of the CoCrFeNiV_0.5_ alloy showed a clearly rising tail, which means that the corrosion was a mass-transfer-controlled process at lower frequency. The radius of the semicircular for the CrFeCoNiV alloy was larger than that for the CrFeCoNiV_0.5_ alloy; this means that the polarization resistance (R_p_) of the CrFeCoNiV alloy was larger than that of the CrFeCoNiV_0.5_ alloy. The solution resistance (R_s_) of the CoCrFeNiV alloy was close to that of the CoCrFeNiV alloy because of the same solution. The polarization resistance (R_p_) and solution resistance (R_s_) values of the CoCrFeNiV_x_ alloys are listed in Table 4. Figure 4c shows the corresponding equivalent circuit diagram of the CoCrFeNiV_x_ alloys in deaerated 1M H_2_SO_4_ solution, where R_s_ is the solution resistance; R_p_ is the polarization resistance; CPE is the constant phase element; and W is the Warburg impedance.

The morphologies of the CoCrFeNiV_0.5_ and CoCrFeNiV alloys after the potentiodynamic polarization tests in deaerated 1M H_2_SO_4_ solution at 30 °C are shown in Figure 5a,b, respectively. The CoCrFeNiV_0.5_ alloy had a granular structure, but the morphology after the potentiodynamic polarization test made it hard to distinguish the grain boundaries, as shown in Figure 5a. The CoCrFeNiV alloy had a dendritic structure, but the morphology after the potentiodynamic polarization test was full of shallow dimples with different sizes, as shown in Figure 5b. The morphology of the CoCrFeNiV alloy also made it hard to distinguish the FCC and σ phases. However, the morphology of the CoCrFeNiV_x_ alloys after the potentiodynamic polarization tests in deaerated 1M H_2_SO_4_ solution showed a uniform corrosion type.

Figure 6a,b shows the potentiodynamic polarization curves of the CrFeCoNiV_0.5_ and CrFeCoNiV alloys tested in deaerated 1M HCl solution at 30 and 60 °C, respectively. The important potentiodynamic polarization data of the curves are listed in Table 5. The corrosion potential of the CrFeCoNiV_0.5_ alloy was more negative than that of the CrFeCoNiV alloy; the corrosion current densities (*i*_corr_) of the CrFeCoNiV_0.5_ alloy were less than those of the CrFeCoNiV alloy at 30 °C, as shown in Figure 6a. Both CrFeCoNiV_0.5_ and CrFeCoNiV alloys had significant anodic peaks. The values of the passivation potential (*E*_pp_) and critical current density (*i*_crit_) of the anodic peak for the CrFeCoNiV_x_ alloy are listed in Table 5. The passivation current density of the CrFeCoNiV_x_ alloys almost kept the same value when the applied potential was less than 0.7 V_SHE_. Then, the secondary anodic peak was observed in each alloy at about 1 V_SHE_. The values of the passivation potential (*E*_pp2_) and critical current density (*i*_crit2_) of the secondary anodic peak of the CrFeCoNiV_x_ alloys are also listed in Table 5. The passivation regions of the CrFeCoNiV_x_ alloys were broken down at a potential (*E*_b_) of about 1.2 V_SHE_ because of the oxygen evolution reaction [22]. The potentiodynamic polarization curves of the CrFeCoNiV_x_ alloys tested in deaerated 1M HCl solution at 60 °C are shown in Figure 6b. The main differences in the CrFeCoNiV_x_ alloys tested at 30 and 60 °C were the passivation regions of the alloys. The passivation current densities of the CrFeCoNiV_x_ alloys increased with the increase in the applied potential. In addition, the secondary anodic peaks of these two alloys vanished at the testing temperature of 60 °C. The passivation region of the CrFeCoNiV_x_ alloys was also broken down at a potential (*E*_b_) of about 1.2 V_SHE_. The corrosion current densities, critical current densities for the anodic peaks of these alloys increased with increasing testing temperature. The corrosion potentials of the CrFeCoNiV_0.5_ alloy were more negative than those of CrFeCoNiV alloy under 30 and 60 °C; the corrosion current densities and passivation current densities of the CrFeCoNiV alloy were larger than those of the CrFeCoNiV_0.5_ alloy. Therefore, the corrosion resistance of the CrFeCoNiV_0.5_ alloy was better than that of the CrFeCoNiV alloy in deaerated 1M HCl solution.

The Nyquist plots, Bode plots and equivalent circuit diagrams of the CoCrFeNiV_x_ alloys tested by electrochemical impedance spectroscopy in deaerated 1M HCl solution are shown in Figure 7. The start points of the CoCrFeNiV_x_ alloys in the Nyquist plot were very close, as shown in Figure 7a. The Nyquist curve of the CoCrFeNiV_0.5_ alloy showed a clearly rising tail, which means that the corrosion belonged to a mass-transfer-controlled process at lower frequency. The radius of the semicircular for the CrFeCoNiV alloy was larger than that for the CrFeCoNiV_0.5_ alloy; this means that the polarization resistance (R_p_) of the CrFeCoNiV alloy was larger than that of the CrFeCoNiV_0.5_ alloy. The differences in solution resistances (R_s_) between these two alloys were small because of the same solution. The polarization resistance (R_p_) and solution resistance (R_s_) values of the CoCrFeNiV_x_ alloys are listed in Table 6. Figure 4c shows the corresponding equivalent circuit diagram of the CoCrFeNiV_x_ alloys in deaerated 1M HCl solution, where R_s_ is the solution resistance; R_p_ is the polarization resistance; CPE is the constant phase element; and W is the Warburg impedance.

Figure 8a,b shows the morphologies of the CoCrFeNiV_0.5_ and CoCrFeNiV alloys after the potentiodynamic polarization tests in deaerated 1M HCl solution at 30 °C. Many shallow dimples were observed on the surface of the CoCrFeNiV_0.5_ alloy after the potentiodynamic polarization test, as shown in Figure 8a. The shallow dimples were distributed on the grain boundaries and within the grains. However, the morphology of the CoCrFeNiV alloy after the potentiodynamic polarization test was full of shallow dimples with different sizes, as shown in Figure 8b. The morphology of the CoCrFeNiV alloy also made it hard to distinguish the eutectic structure, the FCC and σ phases. However, the morphology of the CoCrFeNiV_x_ alloys after the potentiodynamic polarization tests in deaerated 1M HCl solution was a uniform corrosion type.

The corrosion depth of an alloy could be calculated from the relationship between corrosion current density and the charge of the electrons released from the corroded alloy. Under uniform corrosion, the corrosion depth of one year can be calculated from the following equation by assuming that the average density of the alloy is *ρ* = ∑*X_i_ρ_i_*, where *X_i_* and *ρ_i_* are the molar fraction and density of element i:(1)$$\frac{A\cdot D\cdot \rho }{M}\cdot n\cdot F=A\cdot i_{\mathrm{corr}}\cdot t$$
where *A* is the corrosion area; *D* is the corrosion depth of one year; *ρ* is the average density; *M* is the average atomic mass; *n* is the number of average valence electrons; *F* is the Faraday constant (96,500 C/mol); *i*_corr_ is the corrosion current density; and *t* is the corrosion time (3.1536 × 10^7^ s, one year). The corrosion depths of the CrFeCoNiV_x_ alloys per year are listed in Table 7. The results indicated that the corrosion resistance of the CrFeCoNiV_0.5_ alloy was better than that of the CrFeCoNiV alloy in both the deaerated 1M H_2_SO_4_ and 1M HCl solutions. The CrFeCoNiV_0.5_ alloy has a granular structure, and grain boundaries in the alloy were easily to be corroded because of higher Gibbs free energy, as shown in Figure 8a. CrFeCoNiV alloy has a dual-phased dendritic structure; these two phases would form the cathode and anode of local cells in the CrFeCoNiV alloy. Therefore, the corrosion rate of CrFeCoNiV alloy was larger than that of CrFeCoNiV_0.5_ alloy in these two solutions. In addition, increasing the test temperature increased the corrosion rates of the CrFeCoNiV_x_ alloys. However, the results also indicated that the corrosion resistance of the CrFeCoNiV_x_ alloys in deaerated 1M HCl solution was better than that in 1M H_2_SO_4_ solution.

## 4. Conclusions

This work studied the microstructures and corrosion behavior of CrFeCoNiV_x_ high-entropy alloys. The CrFeCoNiV_0.5_ alloy had a granular microstructure, and the CrFeCoNiV alloy had a dendritic microstructure. The two phases existing in these CrFeCoNiV_x_ high-entropy alloys were the FCC and σ phases. The hardness of CrFeCoNiV alloy was much higher than that of CrFeCoNiV_0.5_ alloy, because CrFeCoNiV alloy had a dendritic structure and more σ-phase which was harder than that of FCC phase. The potentiodynamic polarization curves of the CrFeCoNiV_x_ high-entropy alloys indicated that the corrosion resistance of the CrFeCoNiV_0.5_ alloy was better than that of the CrFeCoNiV alloy in deaerated 1M H_2_SO_4_ and 1M HCl solutions. However, both the CrFeCoNiV_0.5_ and CrFeCoNiV high-entropy alloys exhibited good corrosion resistance in these solutions because of the granular structure, especially the CrFeCoNiV_0.5_ alloy in deaerated 1M HCl solutions.

## Figures and Tables

**Figure 1 materials-15-03639-f001:**
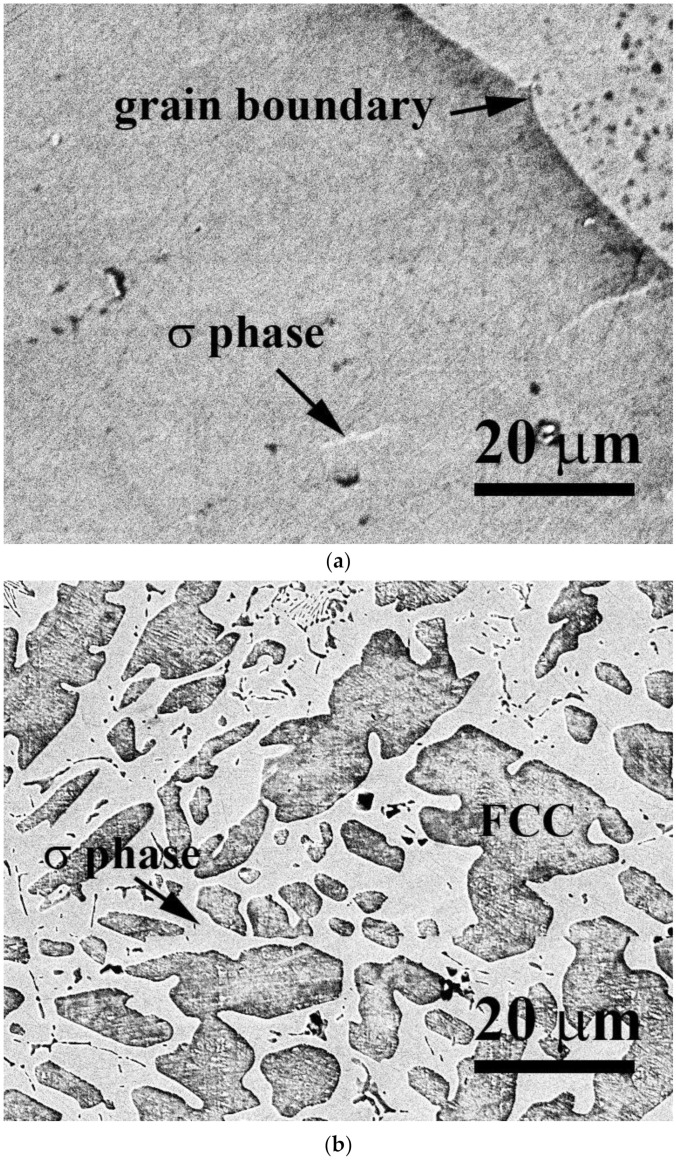
Microstructures of as-cast CrFeCoNiV_x_ alloys: (**a**) CrFeCoNiV_0.5_ alloy and (**b**) CrFeCoNiV alloy.

**Figure 2 materials-15-03639-f002:**
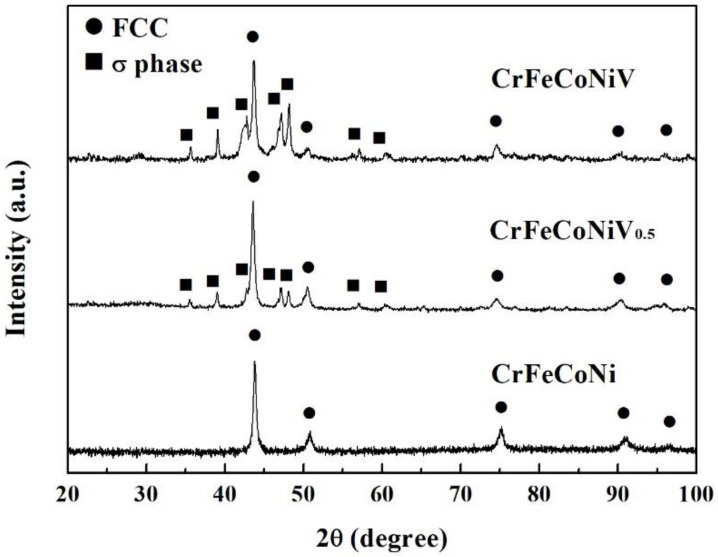
XRD patterns of as-cast CrFeCoNiV_x_ alloys.

**Figure 3 materials-15-03639-f003:**
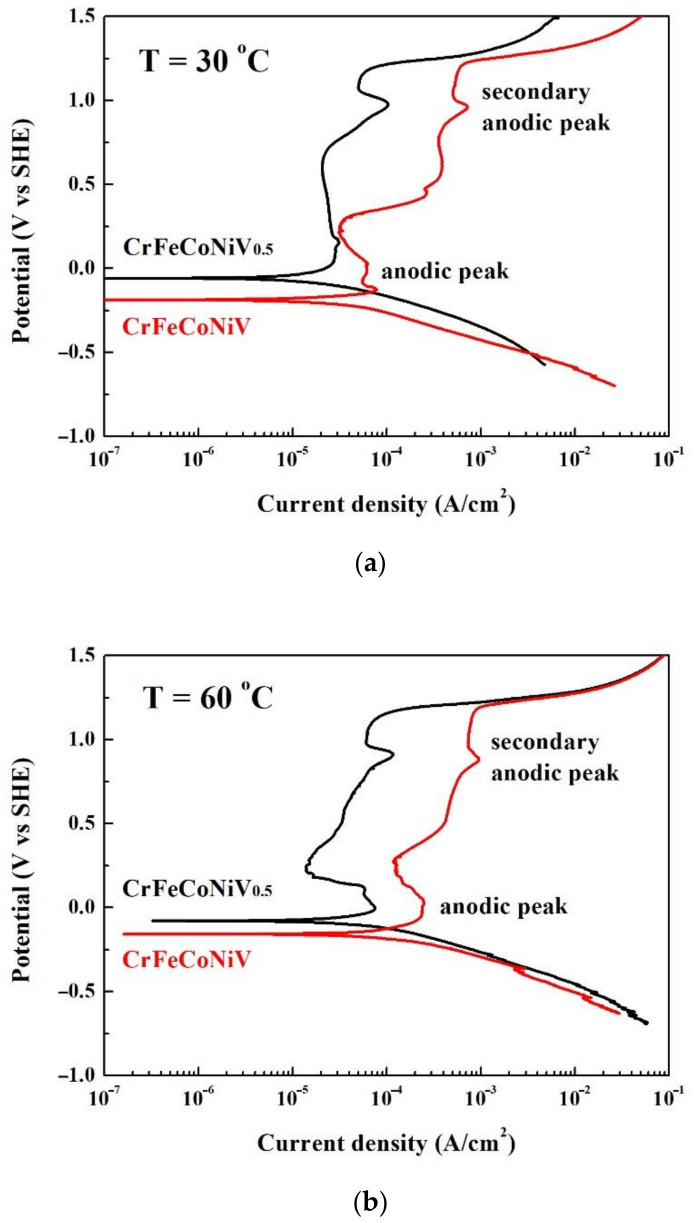
Potentiodynamic polarization curves of as-cast CrFeCoNiV_x_ alloys tested in 1M deaerated H_2_SO_4_ solution at different temperatures: (**a**) 30 °C and (**b**) 60 °C.

**Figure 4 materials-15-03639-f004:**
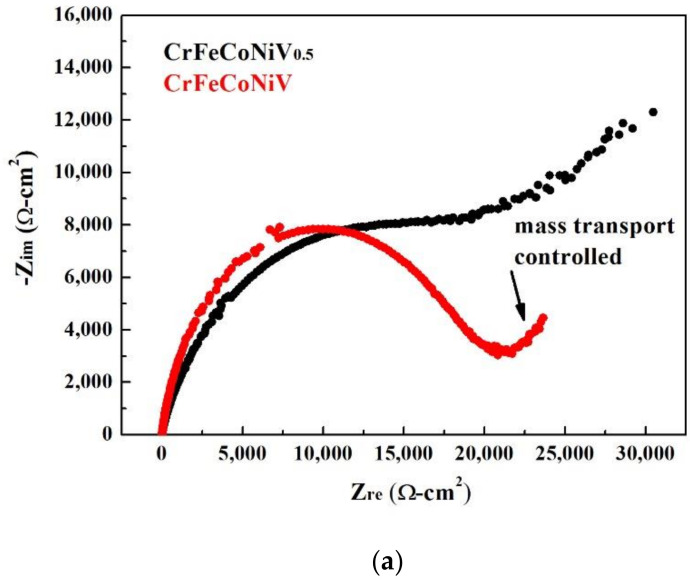
Electrochemical impedance spectroscopy measurements of as-cast CrFeCoNiV_x_ alloys in 1M H_2_SO_4_ solution: (**a**) Nyquist plots; (**b**) Bode plots; (**c**) corresponding equivalent circuit diagrams. R_s_ is the solution resistance; R_p_ is the polarization resistance; CPE is the constant phase element; W is the Warburg impedance.

**Figure 5 materials-15-03639-f005:**
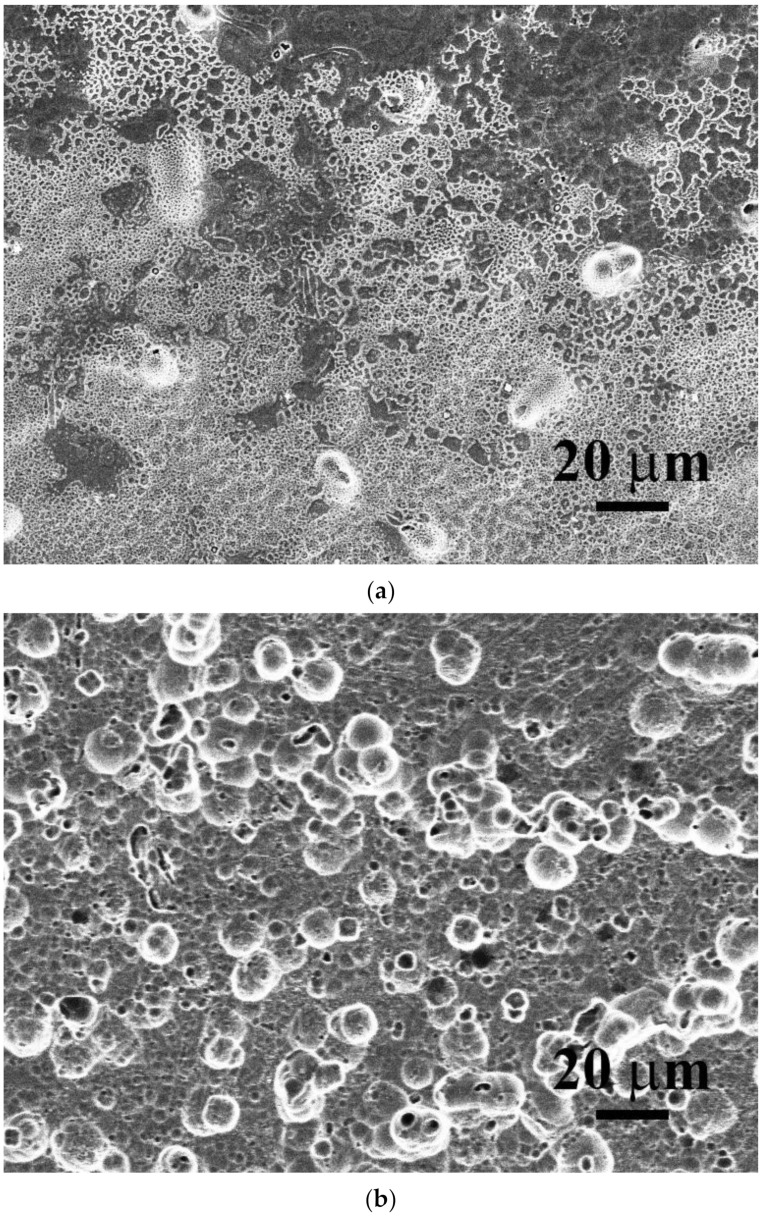
Morphologies of as-cast CoCrFeNiV_x_ alloys after potentiodynamic polarization tests in 1M H_2_SO_4_ solution at 30 °C: (**a**) CoCrFeNiV_0.5_ alloy and (**b**) CoCrFeNiV alloy.

**Figure 6 materials-15-03639-f006:**
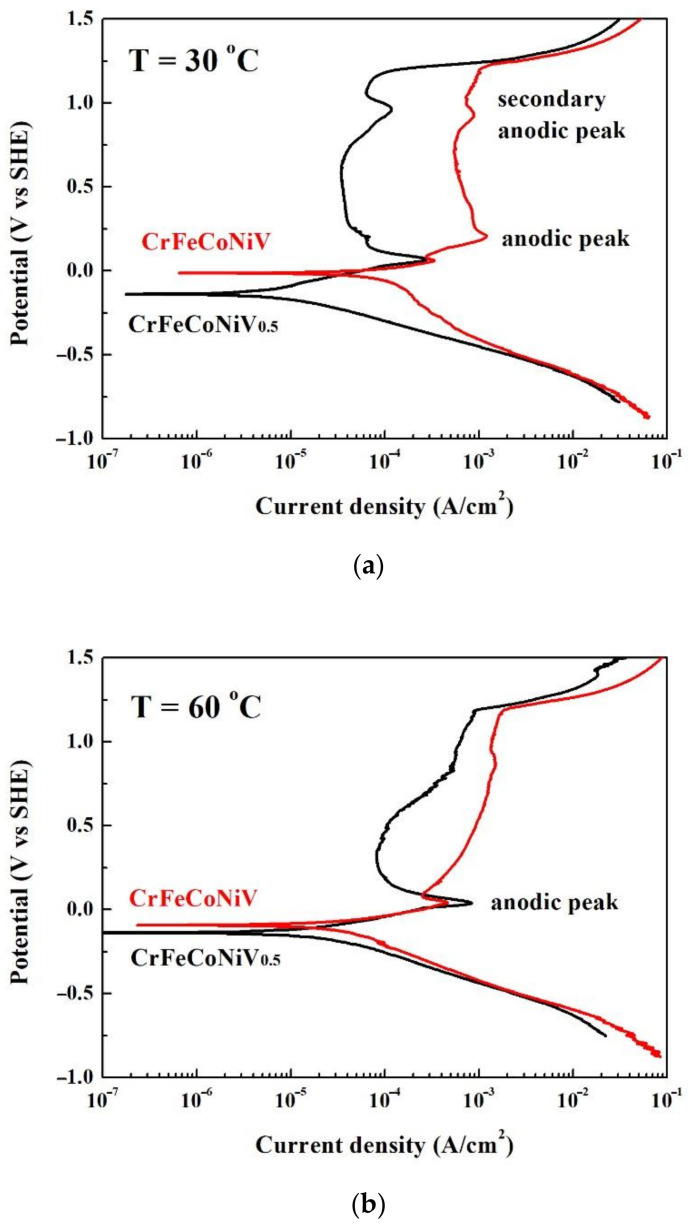
The potentiodynamic polarization curves of as-cast CrFeCoNiV_x_ alloys tested in 1M deaerated HCl solution at different temperatures: (**a**) 30 °C and (**b**) 60 °C.

**Figure 7 materials-15-03639-f007:**
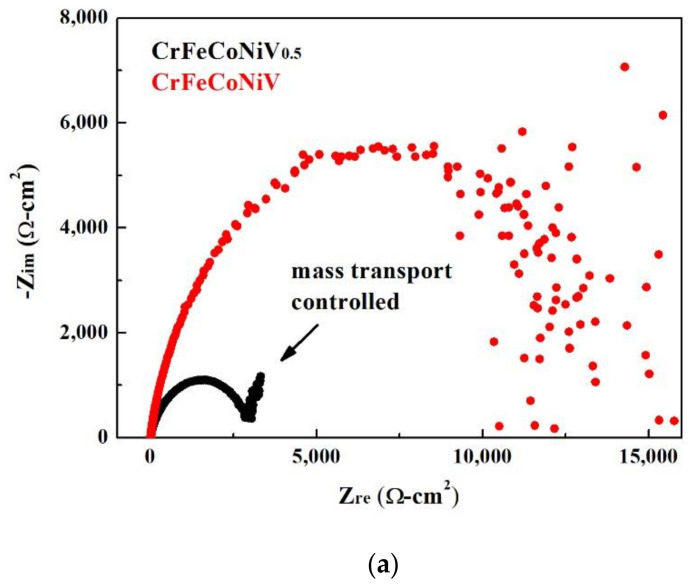
Electrochemical impedance spectroscopy measurements of as-cast CrFeCoNiV_x_ alloys in 1M HCl solution: (**a**) Nyquist plots; (**b**) Bode plots; (**c**) corresponding equivalent circuit diagrams. R_s_ is the solution resistance; R_p_ is the polarization resistance; CPE is the constant phase element; W is the Warburg impedance.

**Figure 8 materials-15-03639-f008:**
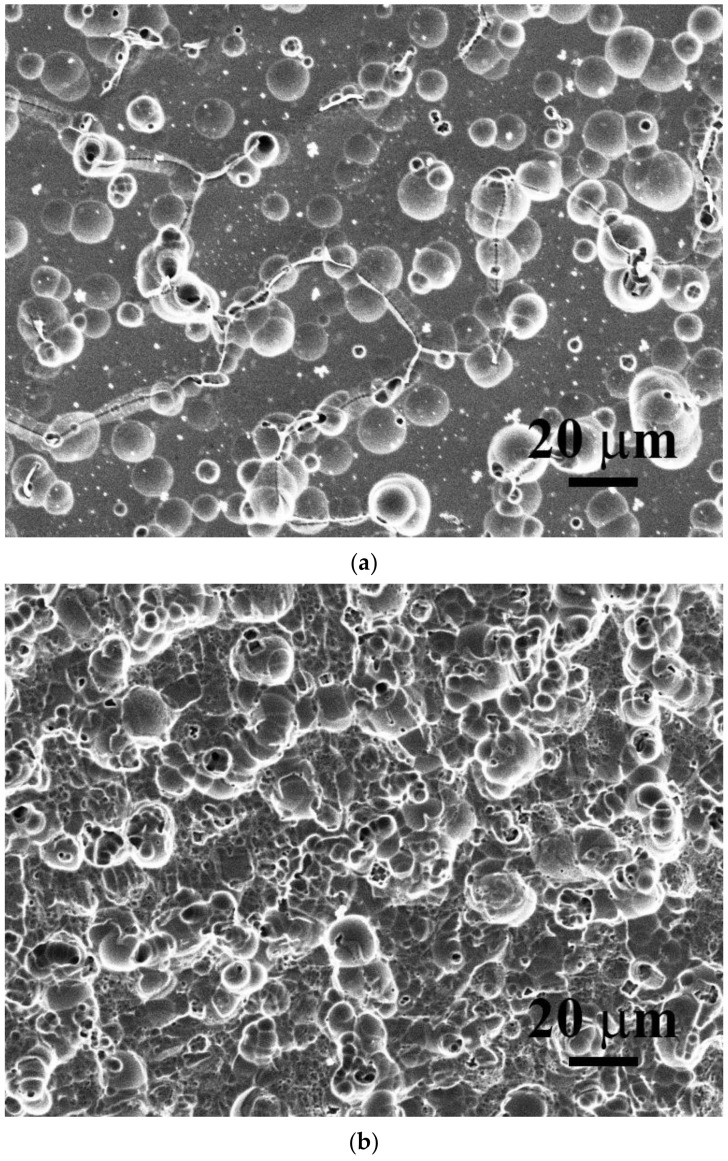
Morphologies of as-cast CoCrFeNiV_x_ alloys after potentiodynamic polarization tests in 1M HCl solution: (**a**) CoCrFeNiV_0.5_ alloy and (**b**) CoCrFeNiV alloy.

**Table 1 materials-15-03639-t001:** Nominal compositions of CrFeCoNiV_x_ alloys.

Alloys (at.%)	Cr	Fe	Co	Ni	V
	(wt.%)				
CrFeCoNiV_0.5_	20.72	22.25	23.48	23.39	10.16
CrFeCoNiV	18.81	20.20	21.32	21.24	18.43

**Table 2 materials-15-03639-t002:** Chemical compositions of as-cast CrFeCoNiV_x_ alloys and the phases in the alloys (analyzed by SEM/EDS).

Regions in the Alloys (at.%)	Cr	Fe	Co	Ni	V
	(wt.%)				
CrFeCoNiV_0.5_					
overall	21.6 ± 0.2	22.6 ± 0.3	22.3 ± 0.4	22.3 ± 0.2	11.2 ± 0.2
FCC	20.9 ± 0.6	23.3 ± 0.7	23.8 ± 0.7	22.7 ± 0.3	9.3 ± 0.9
σ phase	34.1 ± 1.1	20.3 ± 0.2	17.3 ± 0.4	11.7 ± 1.2	16.6 ± 0.3
CrFeCoNiV					
overall	19.4 ± 0.2	20.2 ± 0.2	20.5 ± 0.2	20.2 ± 0.3	19.7 ± 0.2
FCC	17.3 ± 0.6	19.3 ± 0.3	21.1 ± 0.2	23.6 ± 0.8	18.7 ± 0.2
σ phase	23.5 ± 0.2	20.4 ± 0.3	18.9 ± 0.6	15.8 ± 0.2	21.4 ± 0.4

**Table 3 materials-15-03639-t003:** Potentiodynamic polarization data of the as-cast CrFeCoNiV_x_ alloys in 1M deaerated H_2_SO_4_ solution at different temperatures.

	CrFeCoNiV_0.5_		CrFeCoNiV	
	30 °C	60 °C	30 °C	60 °C
*E*_corr_ (V_SHE_)	−0.06	−0.08	−0.19	−0.16
*i*_corr_ (μA/cm^2^)	29.0	69.0	31.0	103
*E*_pp_ (V_SHE_)	0.15	0.00	0.13	0.02
*i*_crit_ (μA/cm^2^)	28.0	75.0	78.0	246
*E*_pp2_ (V_SHE_) *	0.98	0.91	0.96	0.88
*i*_crit2_ (μA/cm^2^) *	104	117	722	962
*E*_b_ (V_SHE_)	1.21	1.18	1.22	1.20

* Secondary anodic peak.

**Table 4 materials-15-03639-t004:** The resistances of as-cast CoCrFeNiV_x_ alloys in deaerated 1M H_2_SO_4_ solution.

	CrFeCoNiV_0.5_	CrFeCoNiV
R_s_ (Ω)	5.62	7.24
R_p_ (kΩ)	5.78	17.1

**Table 5 materials-15-03639-t005:** Potentiodynamic polarization data of as-cast CrFeCoNiV_x_ alloys in 1M deaerated HCl solution at different temperatures.

	CrFeCoNiV_0.5_		CrFeCoNiV	
	30 °C	60 °C	30 °C	60 °C
*E*_corr_ (V_SHE_)	−0.14	−0.14	−0.01	−0.09
*i*_corr_ (μA/cm^2^)	10.0	20.0	20.0	30.0
*E*_pp_ (V_SHE_)	0.07	0.04	0.33	0.04
*i*_crit_ (μA/cm^2^)	28.0	836	1200	463
*E*_pp2_ (V_SHE_) *	0.96	N/A	0.93	N/A
*i*_crit2_ (μA/cm^2^) *	118	N/A	884	N/A
*E*_b_ (V_SHE_)	1.20	1.19	1.23	1.19

* Secondary anodic peak.

**Table 6 materials-15-03639-t006:** The resistances of as-cast CoCrFeNiV_x_ alloys in deaerated 1M HCl solution.

	CrFeCoNiV_0.5_	CrFeCoNiV
R_s_ (Ω)	4.65	7.52
R_p_ (kΩ)	2.91	13.1

**Table 7 materials-15-03639-t007:** Corrosion rates of the as-cast CrFeCoNiV_x_ alloys in deaerated 1M H_2_SO_4_ and 1M HCl solutions.

Alloys	1M H_2_SO_4_ (mm/yr)		1M HCl (mm/yr)	
	30 °C	60 °C	30 °C	60 °C
CrFeCoNiV_0.5_	0.260	0.618	0.090	0.179
CrFeCoNiV	0.257	0.854	0.166	0.249

## Data Availability

Not applicable.

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
