# Peer review of "Corrosion Behavior of CrFeCoNiVx (x = 0.5 and 1) High-Entropy Alloys in 1M Sulfuric Acid and 1M Hydrochloric Acid Solutions"

_materials, 2022, doi:10.3390/ma15103639_

Round 1

Reviewer 1 Report

The "The corrosion behavior of CrFeCoNiVx (x = 0.5 and 1) high-entropy alloys in 1M sulfuric acid and 1M hydrochloric acid solutions" title is a well-written, fluent article. I recommend it for adoption. Please pay attention to the cause of the SI unit system.

Author Response

The "The corrosion behavior of CrFeCoNiVx (x = 0.5 and 1) high-entropy alloys in 1M sulfuric acid and 1M hydrochloric acid solutions" title is a well-written, fluent article. I recommend it for adoption. Please pay attention to the cause of the SI unit system.

Reply: Thank you very much. All of the units have been checked and corrected.

Reviewer 2 Report

The manuscript deals with the influence of V content on the corrosion behaviour of equiatomic and non-equiatomic CoCrFeNiV alloys in two acid media. The study evidences that a decrease in the V content influences significantly the microstructure of the alloys. Thus, the microstructure is FCC, embedding small amounts of sigma phase, for non-equiatomic alloy (low V content) with a complex multiphase dendritic structure for the equiatomic alloy (high V content). The corrosion behaviour has been evaluated through polarization and impedance tests. Electrochemical data calculated from these experiments are provided but there is no correlation between the corrosion rates calculated and the corrosion scale as well regarding the influence of the microstructure on the nature of the corrosion scale. The aspect of polarization curves for both alloys is almost identical although icorr is somewhat higher for the equiatomic alloy. The manuscript does not go deep in the corrosion scale formed on both alloys, and it is not discussed how the microstructure of the alloy influences the corrosion process. It is obvious that the multiphase microstructure of the equiatomic alloy is more propitious for corrosion that the almost single-phase microstructure of the non-equiatomic alloy. Such discussion should be taken into account. The manuscript should be modified considering such aspects before it can be suitable for publishing

Other points to be considered.

  • Why the corrosion potential for the non-equiatomic alloy is higher than that of the equiatomic alloy in sulfuric acid and the opposite is true in HCl?
  • There is not good agreement between icorr values and Rp values in both media. Corrosion rate is lower for non-equiatomic alloy but Rp is higher for the equiatomic alloy. How can it be possible? Are the authors confident about the circuit used for modelling impedance measurements?
  • The surface aspect of the samples subject to polarization curves does not provide relevant information about the corrosion behaviour of the alloy because at the end of the curves the material undergoes polarization phenomena resulting from the high potential applied which re not representative of the corrosion behaviour. It would be more interesting compare the surface images of the corrosion scale in the passivation region.
  • What is the reason of the double peak found in the passivation region?

Author Response

  • Why the corrosion potential for the non-equiatomic alloy is higher than that of the equiatomic alloy in sulfuric acid and the opposite is true in HCl?

Reply: The corrosion potential goes positive or negative depends on the corrosion behavior. Corrosion potential can be calculated by Nernst equation, but the concentrations of ions in the solutions must be measured. Different corrosion results would have different effects on the corrosion potential. We did not measure the concentrations of ions in the solutions, and we cannot calculate the corrosion potential by Nernst equation. This is not included in this manuscript.

  • There is not good agreement between icorr values and Rp values in both media. Corrosion rate is lower for non-equiatomic alloy but Rp is higher for the equiatomic alloy. How can it be possible? Are the authors confident about the circuit used for modelling impedance measurements?

Reply: These two alloys are different alloy systems, although they have same elements, but V-content is different. CrFeCoNiV0.5 is soft granular structure and CrFeCoNiV is hard dendritic structure; they are two alloy systems. Therefore, the corrosion behavior are different.

  • The surface aspect of the samples subject to polarization curves does not provide relevant information about the corrosion behaviour of the alloy because at the end of the curves the material undergoes polarization phenomena resulting from the high potential applied which re not representative of the corrosion behaviour. It would be more interesting compare the surface images of the corrosion scale in the passivation region.

Reply: We can understand the corrosion type of these alloys in these solutions are a uniform corrosion type from the morphologies of the alloys after polarization test. Also, we can calculate the corrosion rates of the alloys after confirming the uniform corrosion type. The surface images of the alloys in passivation region are not included in this manuscript, because it is hard to understand the corrosion behavior from the slight corroded surfaces.

  • What is the reason of the double peak found in the passivation region?

Reply: According to the literature (Clayton, C.R.; Lu, Y.C. A Bipolar Model of the Passivity of Stainless Steel: The Role of Mo Addition, J. Electrochem. Soc. 1986, 133, 2465-2473.), the secondary anodic peak is due to the formation of iron, cobalt and nickel oxyhydroxides. This is not included in this manuscript because the concentrations of ions in the solutions were not measured.

Reviewer 3 Report

The paper ”Corrosion behavior of CrFeCoNiVx (x = 0.5 and 1) high-entropy alloys in 1M sulfuric acid and 1M hydrochloric acid solutions” is suitable for publication in Materials Journal after some minor corrections.

The authors should update the introduction with the influence of the alloying elements over microstructure, mechanical and corrosion properties.

Figure 1 a) should be replaced with a  newer one. It has scratches.

Please update in text with ICDD codes for XRD analysis (data for figure 2).

There are data errors for table 2, but for tables 3-6, there are not. Please update.

There are 9 references below 2013. Please update with newer ones.

Author Response

The authors should update the introduction with the influence of the alloying elements over microstructure, mechanical and corrosion properties.

Reply: We have described the corrosion properties of CrFeCoNiX alloys, and purpose of adding vanadium into CrFeCoNi alloy. These are in the “Introduction”.

Figure 1 a) should be replaced with a newer one. It has scratches.

Reply: Figure 1 (a) and (b) are modified. The portion with scratches has been cut.

Please update in text with ICDD codes for XRD analysis (data for figure 2).

Reply: The code has been added into the text, line 97.

There are data errors for table 2, but for tables 3-6, there are not. Please update.

Reply: The compositions in Table 2 are averaged from three or four tests. The values in Tables 3-6 are form corresponding figures, the data in the Table are not average values.

There are 9 references below 2013. Please update with newer ones.

Reply: These references, 7 articles and 2 books, are important to our manuscript. We also referred the articles published recently.

Round 2

Reviewer 2 Report

Tyhe articles have not addressed the issues marked by this reviewer and some of them are enough for rejecting the manuscript in the present state. Basically, there is no correlation between the microstructure and the corrosion behaviour of the alloys. According to this reviewer, this point should explain the different corrosion behaviour of the alloys. The single -phase or two-phase dendritic structure should influence the corrosion resistance of the alloy. Moreover, it is not discussed how the change in the content of the different elements, especially in the case of vanadium, influences the nature of the corrosion scale, which should determine the overall corrosion behaviour of the alloys . Also it is wrong that the alloy exhibiting the highest icorr presents the lower Rp. The opposite should be true.  The figures presenting the surface of the corroded alloys after polarization tests do not provide useful information about the corrosion process, so they should be removed.   

Author Response

Tyhe articles have not addressed the issues marked by this reviewer and some of them are enough for rejecting the manuscript in the present state.

Basically, there is no correlation between the microstructure and the corrosion behaviour of the alloys. According to this reviewer, this point should explain the different corrosion behaviour of the alloys.

Reply: The figures of the alloys after polarization test showed the correlation between the microstructures and the corrosion behavior of the alloys, so these figures must be kept. After polarization test in the H2SO4 solution at 30 C, both of these two alloys showed severely corroded surfaces, and hard to distinguish the effects of granular or dendritic structures. The corrosion rate of these alloys are thus the corrosion rates of these alloys are very close. Otherwise, after polarization test in the HCl solution at 30 C, CrFeCoNiV0.5 alloy showed a slightly corroded morphology, and it also showed the where the easily corroded area were (the grain boundaries); and CrFeCoNiV alloy showed a severely corroded morphology. This could prove that the corrosion rate of CrFeCoNiV0.5 alloy was lower than that of CrFeCoNiV alloy. This is the difference between the alloys with a granular structure or a dendritic structure, and the dendritic-structured CrFeCoNiV alloy has much interface which is easily corroded. The corroded surfaces of the alloys are important.

The single -phase or two-phase dendritic structure should influence the corrosion resistance of the alloy. Moreover, it is not discussed how the change in the content of the different elements, especially in the case of vanadium, influences the nature of the corrosion scale, which should determine the overall corrosion behaviour of the alloys.

Reply: The vanadium-content influenced the microstructures of the alloys, and also influenced the corrosion behavior of the alloys. This has been described in the manuscript. The corrosion scales of the alloys could be understand from the images of the alloys after polarization test.

Also it is wrong that the alloy exhibiting the highest icorr presents the lower Rp. The opposite should be true.

Reply: CrFeCoNiV0.5 and CrFeCoNiV are two different alloys, they have different microstructures, one is granular structure and another is dendritic structure. Therefore, the icorr and Rp of these alloy cannot be compared. However, the relationship of icorr and Rp of same alloy in these two solutions can answer this question. The icorr and Rp of CrFeCoNiV0.5 alloy are 29 uA/cm2 and 5.78kohm in H2SO4 solution, respectively; and those are 10uA/cm2 and 2.91 kohm in HCl solution, respectively. Also, the icorr and Rp of CrFeCoNiV alloy are 31 uA/cm2 and 17.1kohm in H2SO4 solution, respectively; and those are 20uA/cm2 and 13.1 kohm in HCl solution, respectively. All of these quite fit the question of reviewer. It is important that CrFeCoNiV0.5 and CrFeCoNiV are two different alloys, they have different structures.

The figures presenting the surface of the corroded alloys after polarization tests do not provide useful information about the corrosion process, so they should be removed.

Reply: These figure could prove the uniform corrosion type of these alloys, and they are important in this manuscript as described above. So that, we still kept these figures.

Round 3

Reviewer 2 Report

The authors have not introduced any kind of improvement in the manuscript. Furthermore, the arguments used to reply the comments of this reviewer have no sense. The authors claim a correlation between the microstructure of the alloys and the corrosion behaviour based on the morphology of sample surfaces after polarization tests. In polarization tests the potential is gradually increased, moving so far from the equilibrium conditions. Consequently, surface images of the corroded samples are not representative of the corrosion process. In fact, extensive pitting is observed in both materials as a consequence of the high potential appied to the material. It would be more illustrative to present the images of the scale for different exposure times in open circuit conditions or in polarization tests in a region not so far from the corrosion potential. It is not discussed why the corrosion rate of dendritic alloy is higher than that of the single-phase alloy. It is obvious that in the dendritic alloy one phase will act as cathodes and the other phase as anodes, exacerbating in this way the corrosion process, but nothing is said about if throughout the manuscript. Both alloys can develop a passive layer. There are no information about the nature of the corrosion scale (XRD, XPS or EDS data). There is no information about the changes induced in the composition of the scale as result of the decrease in the content of vanadium. It seems that the nature of the corrosion products formed on both alloys is the same because the shape of passivation stage in polarization curves is similar, but this should be confirmed. Finally, there is no good agreement between the icorr calculated from polarization tests and Rp values calculated from impedance tests.  Higher current densities (icorr) imply a less protective corrosion scale, i.e. Rp should be also lower. Nevertheless, the alloy with higher current densities (equiatomic alloy) exhibits the highest Rp. This means that the corrosion scale formed on the equiatomic alloy is more protective than that developed on the non-equiatomic alloy, but the opposite it is true according to polarization tests. How can the authors explain such inconsistency?